# Assessing the Effects of Landscape Change on the Occupancy Dynamics of the Greater White-Toothed Shrew *Crocidura russula*

**DOI:** 10.3390/life12081230

**Published:** 2022-08-14

**Authors:** Ignasi Torre, Mario Díaz

**Affiliations:** 1BiBio Research Group, Natural Sciences Museum of Granollers, C/Francesc Macià 51, E-08402 Granollers, Spain; 2Department of Biogeography and Global Change (BGC-MNCN-CSIC), National Museum of Natural Sciences, C/Serrano 115 Bis, E-28006 Madrid, Spain

**Keywords:** *C. russula*, landscape change, occupancy models, spatial distribution, scrubland encroachment, afforestation, small mammals, shrews

## Abstract

Land-use change is the main driver of biodiversity loss in the Mediterranean basin. New socio-economic conditions produced a rewilding process so that cultural landscapes are being invaded by more natural habitats. We analyze the effects of landscape change on the demography and the spatial distribution of *Crocidura russula* in six protected areas of the western Mediterranean basin. The study was conducted in the period 2008–2020 on 19 live trapping plots representing the three main natural habitats of the area (scrubland, pinewood, and holm oak woodland). We used a multiscale approach to ensure that the scale of response matched landscape structure (from plot to landscape) using either vegetation profiles (LiDAR) and land use data obtained from years 2007 and 2017. Statistical models (multiple-season single-species occupancy models) showed that *C. russula* populations were strongly associated to habitat features at the plot level. These models were used to predict occupancy at sampling units for the whole study area (850 km^2^), showing contrasting trends that shifted at relatively small spatial scales (expansions and retractions of species ranges). Parks showing extreme scrubland encroachment (−8% of area) and afforestation (+6%) significantly reduced habitat suitability for shrews and reductions in occupancy (−5%). Results would indicate faster changes in the spatial distribution of the target species than previously expected on the basis of climate change, driven by fast landscape changes.

## 1. Introduction

Land-use change (and associated pressures) is the main driver of biodiversity loss worldwide [1]. Land use and land cover change is a cause of the current biodiversity crisis, but its relationship with biodiversity conservation remains largely unknown, especially at the regional scale due to the lack of consistent biodiversity data [2]. Site- and habitat- based analyses of available long-term datasets for common species would greatly contribute to uncover this gap of knowledge [3].

In the Mediterranean basin, complex interactions between ecosystems and humans resulted in the regression of natural habitats (forest) through shifting land use practices extended over millennia [4]. Despite that habitat conversion for human uses (e.g., agriculture and urban areas) is characteristic of the Anthropocene [5], more recently, abandonment of traditional uses has been followed by a partial regrowth of natural vegetation and recovery of wildlife diversity and abundance [6,7]. Effects of landscape dynamics (land use intensification or abandonment) on biodiversity is key for realistic and effective planning of landscape management actions aimed at achieving specific conservation goals (e.g., [8]).

Alteration of the landscape caused by land use practices or climate change can affect habitat suitability for many species, producing impacts in demography affecting colonization and extinction probabilities. At the population level, those changes manifest as shifts in the occurrence and abundance of species across the landscape [9]. Populations can be affected by the habitat loss within a region and by changes in the suitability of fragmented remains of habitat regarding the isolation or connectedness of patches lasting after habitat loss [10]. Understanding how these dynamic processes are affected by changes of habitat or climatic conditions may be important for the successful management of ecological systems. For example, in metapopulation studies, local extinction probabilities are frequently assumed to be decreasing functions of patch area (e.g., [11]). In this regard, it is important to effectively define the scales at which responses of the target species are expected to occur, and multiscale approaches are recommended [5].

Here, we analyse the effects of landscape change on the demography and spatial distribution of a small endotherm, the greater, white-toothed shrew *Crocidura russula*, in protected areas of the western Mediterranean basin. This is a thermophilic species with a Mediterranean distribution, showing clear-cut open-land habitat preferences, and can be considered as a suitable target species for demographic studies owing to the short life span, high dispersal ability, small ranges, and high detectability [12,13]. Previous work showed that, contrary to expectations, the abundance and demographic parameters of these shrew populations were not (yet) affected by ongoing climate change, although they strongly differed among contrasting habitat types (open woodland, [14]). We expand the scale of analyses by including effects of landscape change at regional scales after accounting for local-scale effects of vegetation structure [5]. Models including land uses at landscape scales, that are typically measured using GIS-based assessments updated by public administrations [15], may allow, if significant, to create maps of the likelihood of species’ occurrence at places that will not be surveyed, thus enlarging the spatial scope of results at the regional scales relevant for land-use policy evaluations [16].

## 2. Materials and Methods

### 2.1. Study Area

Field work was conducted within six natural parks of the Barcelona province (Catalonia, NE Spain, Figure 1). The study area is very heterogeneous regarding landscape composition, with large areas occupied by open habitats in the south (Garraf and Olèrdola) and mostly forest-covered areas in the north (Montnegre i el Corredor). This area—as well as the entire region—is under a general process of shrub encroachment and afforestation during the last five decades [17,18]. Woodlands are dominant (57%), followed by open natural habitats (shrublands and grasslands, 27%), mostly originated by wildfires at the end of the last century and the start of the present one. Non-natural habitats, such as urban areas (11%) and croplands (4%), showed a reduced area and were mostly situated in the periphery of the natural parks [2,19].

### 2.2. Small Mammal Sampling

We used the database of the SEMICE small mammal monitoring program (www.semice.org, accessed on 15 February 2022). It is a mixed monitoring scheme with stations operated by both professionals and volunteers, whose main goal is monitoring common small mammal species with high detectability [20].

The study plots were selected according to the available SEMICE stations that represents a non-random sample of natural Mediterranean habitats representative of the study area. The sampling period included 13 years, from spring 2008 to fall 2020, and 19 live trapping plots of 0.56 ha (minimum area) were monitored twice a year (spring and autumn) to cover the phases of the life cycle of the species [21,22]. Open habitats included post-fire vegetation communities dominated by scrublands of *Quercus coccifera*, *Pistacia lentiscus*, and *Cistus* spp., and woodlands included a variety of evergreen (*Quercus ilex* with *Pinus pinea* and *P. halepensis*) and deciduous (*Quercus pubescens*, *Alnus glutinosa*, and *Salix* spp.) forests. Our investigation was centered on the lowlands (95–750 m a.s.l.) to prevent the influence of climate variability on species diversity and abundance [23]. Sampling plots consisted of grids of 36 traps spaced 15 m (6 × 6 and 9 × 6 scheme design), and to avoid size-specific biases in small mammal community assessments [24], we used two types of live traps: Longworth (Longworth Scientific Instrument Co., Oxford, UK) and Sherman traps (Sherman folding small animal trap; 23 × 7.5 × 9 cm; Sherman Co., Tallahassee, FL, USA), that were alternated in position [25,26]. Traps were baited with a piece of apple and a mixed dough of tuna and flour and provided with hydrophobic cotton for bedding [27]. Traps were operated during three consecutive nights and revised during the early morning of the first, second, and third day. Shrews were identified, weighed, marked with fur clips to detect short-term recaptures, and released at the point of capture [28]. Research on live animals followed ethical guidelines [29], and captures were performed under special permission of the Catalan Government (Generalitat de Catalunya).

### 2.3. Vegetation Structure and Land-Use

Vegetation structure of sampling plots was assessed by ALS LiDAR [30,31] (Figure 2) obtained from the Institut Cartogràfic i Geològic de Catalunya (flights 2016–2017). This method is adequate for analysing small mammal-vegetation relationships [32,33]. Height of vegetation is relevant for small mammals’ microhabitat selection [34,35], and 12 variables describing vertical vegetation structure and height were derived at the plot level (Appendix A). Redundant variables describing horizontal profiles of vegetation (e.g., cover) were disregarded, owing to the positive relationship between height and cover [36].

To study the effects of vegetation structure and land use on *C. russula* populations, we used a multiscale approach to ensure that the scale of response matched landscape structure [5] (Figure 3): (1) Analyses of the effects of vegetation structure at the plot level, since microhabitat features were expected to be relevant drivers for initial occupancy. (2) In the context of meta-population dynamics [11], some population parameters (i.e., local colonization and extinction) are expected to be determined by the landscape matrix surrounding the plots; hence, we analyzed the relationships between landscape structure and changes in occupancy and derived parameters, considering two circular buffers of 100 (plot) and 500m radius (landscape) centered on the plots. Owing to the spatial patterns of land use and cover change at the landscape scale in the study area [2], we expected consistent patterns of landscape change at different spatial scales (plot and landscape, [5] Figure 3). (3) Spatial distribution models were projected using a geographic information system (GIS) to a 1 km^2^ land-use database of 2007 and 2017 available for the entire area (850 km^2^) [15]. The area projected does not conform exactly to the boundaries of the natural parks, since we considered all the 1 × 1 km units which were crossed by or included within the limits of the protected areas.

Since small mammals are likely to respond to complex combinations of habitat components [37], we obtained gradients of either vegetation structure (LiDAR, at the plot level) and land-use (at landscape level). Principal component analyses (PCAs) were performed, including the 12 LiDAR variables (step 1), on the four main land uses (scrublands, forests, crops, and urban) around sampling stations (500 m buffer, step 2), and the other two on the same land-use variables at the start (2007) and the end (2017) of the study (step 3). Components were rotated using the Varimax method, and only those having eigenvalues >1 were retained (Kaiser criterion). These new variables can be interpreted as gradients with ecological meaning and were used as predictors in further analyses [38]. Land use change at the plot level was analyzed calculating the rate of change of the main natural (scrubland and forest) and anthropic habitats (crops and urban areas) between the two periods (2007–2017) at two spatial scales: 100 mbuffer (plot) and 500 mbuffer (landscape). To ascertain whether changes were consistent between the two periods, Wilcoxon matched-pairs tests were used. Land use change at the widest scale (entire region, natural parks) was analyzed with generalized linear models (GLMs), using the rate of change of the four mainland uses as response variables and the natural parks as predictors.

### 2.4. Occupancy Analysis

Multiple-season single-species occupancy models [11] were used to analyze the range dynamics of *C. russula*. This approach allows modelling two separate processes following a hierarchical structure [39]: the “state process” analyzing the species distribution and its drivers, and a process associated to the data collection (“observation process”). The timeframe database considers two temporal scales: a short one, consisting in the detection/non-detection of the species during the three consecutive trapping sessions conducted during a survey (secondary occasions) to account for species detectability; a long one, consisting in the detection/non-detection of the species during consecutive seasonal surveys (primary occasions) to account for changes in occupancy and the rest of parameters. This approach can be used when the assumption of closure—no local immigration or emigration—between primary occasions is potentially violated, but where closure can be assumed within each primary occasion [40]. These kinds of models allow first-order Markovian changes in occupancy, that is, when occupancy at a site in the present season depends on the state of occupancy at that site in the last season [11]. Models calculate parameters such as local colonization and local extinction probabilities to account for changes in occupancy between seasons, being that occupancy (*ψ*) is the proportion of sites that is occupied by the target species; colonization (*γ*) is the probability that an unoccupied site in season t is occupied in season t + 1; extinction (*ε*) is the probability that a site occupied in season t is unoccupied in season t + 1; persistence (*Ø*) is the probability of a site being occupied in successive seasons (1 − *ε*); and detectability (*p*) is the probability of detecting the species when actually present. The SEMICE monitoring program is especially suitable for applying such a kind of statistical models, since it consists in three surveys repeated in three consecutive days for every sampling site, and surveys are repeated in two annual seasons along the years. Furthermore, *C. russula* showed high detection probabilities under the SEMICE programme [14,20], well above the *p* = 0.3 threshold indicating high likelihoods for false negatives [41], and cumulative detectability was also high (*p* > 0.85) for all the surveyed sites [39] and references therein.

We fitted competing occupancy models (software PRESENCE, [16]) to determine the parameters (occupancy, colonization, and extinction probabilities) affecting the population dynamics of the species at different spatial scales, once controlling for imperfect detectability. We started with a null model, considering that occupancy rate did not change in space and time and setting constant the parameters colonization, extinction, and detection probabilities [*ψ(·)γ(·)ε(·)p(·)*]. Regarding the lack of automatic selection of best models in the software used, a limited set of covariates was included in models by using sound scientific judgment around the target species [11]. Hence, the models were improved adding vegetation structure and land use profiles of plots (site-dependent covariates) and seasonal effects and considering them altogether in the models but without interactions [42]. Owing to the well-known habitat preferences of the species in the area [14,43], we expected that initial probability of occupancy at the plot level to be mainly determined by vegetation structure profiles of sampling plots (LiDAR profiles). However, we expected that other parameters (colonization, extinction, and persistence) to be also constrained by temporal changes in suitability of habitats occurring around sampling stations; thus, we included land use change (i.e., the rate of change of open and forest land cover between 2007 and 2017) as site-dependent covariates at the two spatial scales (100 and 500 m buffers). Detectability strongly depends on habitat features [42] and was expected to change according to vegetation profiles (LiDAR) of plots, but also on a seasonal basis, owing to marked seasonality of species abundance [15,36]. Finally, we projected the probability of occupancy at the scale of the natural parks network in both periods, considering the modelled occupancy calculated in 2007 and 2017: [*ψpc*_2007_*, γ(pc*_2007_*), ε(pc*_2007_*), p(t + pc*_2007_*)*], and the same for 2017.

## 3. Results

We trapped 3975 small mammals of seven species along the study period (2008–2020). *Apodemus sylvaticus* was dominant (61%), followed by *C. russula* with 986 individuals (25%) and *Mus spretus* (11.4%). The other four species reached less than 2% of captures each. A total of 61% of shrews were captured during the autumn surveys. Shrew abundance was higher in scrubland plots (7.49 ± 5.39 ind./plot; mean ± SD) than in forests plots (1.42 ± 2.25 ind./plot).

### 3.1. Patterns of Land Use Change

Afforestation and loss of scrubland involved 67% of the study area, with 3480 ha affected by scrub encroachment and 2650 ha by afforestation. The opposite land use change affected the remaining 23% of the area, where scrubland increased by 1380 ha and open deforested lands by 1750 ha. Overall, the region showed a reduction of scrubland (−2.66%) and an expansion of forests (+1.32%), crops (+0.67%), and urban (+0.66%) areas (Table 1). The pattern differed among the natural parks sampled. Serralada Litoral experienced the strongest landscape change affecting 91% of the area, with a loss of 678 ha of scrubland (−8.6%) and an increase of 481 ha of afforestation (+5.9%). Collserola was the only park where no scrubland loss occurred, but an increase in the crop cover occurred (0.26%, 45 ha). A similar pattern was observed in Sant Llorenç del Munt i l’Obac, with a small reduction of scrubland (−0.2%, 60 ha) and an expansion of land devoted to crops (+0.62%, 113 ha). Both parks showed a net increase of the surface of open habitats due to recuperation of crops and a reduction of the loss of scrubland. Both parks also showed a reduction of the forest cover.

A generalized linear model with the rate of change of land use by natural park yielded significant results for scrubland (Wald Chi^2^ = 71.1, df = 5, *p* < 0.0001), forest (Wald Chi^2^ = 53.2, df = 5, *p* < 0.0001), crops (Wald Chi^2^ = 12.6, df = 5, *p* < 0.03), and urban (Wald Chi^2^ = 112.0, df = 5, *p* < 0.0001). The only land use increasing in all the parks was urban cover, but crop cover only increased in S. Litoral. Forest cover increased in three parks (Montnegre, S. Litoral, and Garraf), and scrubland cover decreased in the same parks.

### 3.2. Vegetation Structure and Land Use of the Sampling Stations

The PCA performed with the 12 LiDAR variables resulted in four significant factors (with eigenvalues >1). The first two PCs accounted for 76.13% of variance (54.35% for PC1 and 21.78% for PC2, Appendix A) and were considered for further analyses. PC1 was correlated with nine variables, and segregated the scrublands from the woodlands, being interpreted as a gradient of increasing complexity of vertical vegetation structure. PC2 correlated with four variables and was mostly associated to vegetation cover and height of short and tall shrubs. Scrublands showed high vegetation cover of short shrubs (0.15–1.50 m tall), low vegetation cover of tall vegetation (>2.50 m), and a lack of canopy cover. At the other extreme, broad-leaved woodlands showed low vegetation cover of short shrubs and high vegetation cover of the tall vegetation.

Factor coordinates of LiDAR-PC1 were significantly correlated with land uses at the plot (100 m buffer) and the landscape scales (500 m buffer) in both periods and for the two main habitats: LiDAR-Openland 2007: *r* = −0.96 and −0.79; LiDAR-Openland 2017: *r* = −0.95 and −0.79 and LiDAR-Woodland 2007: *r* = 0.96 and 0.78; LiDAR-Woodland 2007: *r* = 0.95 and 0.78. Therefore, vegetation structure profiles summarized by LiDAR-PC1 offered equivalent information than land use composition at the plot and landscape scales.

At the start of the study (2007) and considering the landscape scale (500 m buffer), land use around sampling plots was dominated by forest (62.2% ± 35.1 SD) followed by open land (33.6% ± 34.6 SD), but at the end of the study (2017), forest cover had increased (65.2% ± 34.1 SD) and open lands had decreased in extent (29.8% ± 33.2 SD). In fact, open natural habitats (grassland and scrubland) decreased 4.5% on average (*z* = 3.00, *p* = 0.002, *n* = 19) and forest increased 3.1% (*z* = 2.77, *p* = 0.005, *n* = 19), whereas crops and urban areas mostly remained unchanged (<1% change). These patterns were almost identical in all sampling plots, suggesting that landscape change was consistent irrespective of the composition of habitats surrounding the plots. Indeed, 14 out of 19 plots (74%) showed scrubland regression and afforestation, 4 plots showed no change, and only 1 plot showed a reversed pattern (increase of scrubland and decrease of forest, Figure 4). Furthermore, land use rate of change at the plot level showed negative correlation between both main natural habitats (scrubland-woodland, *r* = −0.93, *p* < 0.001, *n* = 19), suggesting that scrubland regression was mostly associated to afforestation around the studied plots. However, land use changes in the same period were non-significant at the plot scale (100 m buffer): forest (*z* = 0.56, *p* = 0.57, *n* = 19) and open land (*z* = 0.91, *p* = 0.36, *n* = 19), due to high heterogenous and inconsistent land use patterns of change within plots (Figure 4). In fact, nine plots showed no change (47%), four showed forest regression and scrubland expansion, three showed scrubland regression and afforestation, two showed only afforestation, and one showed afforestation and scrubland expansion.

### 3.3. Occupancy and Spatial Distribution Models

At the plot scale, the most parsimonious occupancy model (AIC weight 100%) included LiDAR structure for initial occupancy, colonization, extinction, and detectability (Figure 5) and landscape change (scrubland and forest rate of change in the period 2007–2017) for colonization and extinction:*ψ(LiDAR), γ(LiDAR + Scrub100 + Forest100), ε(LiDAR + Scrub100 + Forest100), p(season + LiDAR).*

At the landscape, the most parsimonious occupancy model (AIC weight 76%) included LiDAR structure for initial occupancy, colonization, extinction, and detectability and landscape change (scrubland and forest rate of change in the period 2007–2017) for colonization and extinction:*ψ(LiDAR), γ(LiDAR + Scrub500 + Forest500), ε(LiDAR + Scrub500 + Forest500), p(season + LiDAR).*

Despite that occupancy was constant along the study period, a predicted decline of occupancy was obtained for the two pinewood plots, due to decreasing colonization probabilities associated to landscape change around the plots (high rates of scrubland reduction and forest expansion). During the first period (spring 2008–spring 2013), the average probability of occupancy was 0.67 ± 0.18 SE, and during the second period (autumn 2013–autumn 2020), the occupancy declined (0.48 ± 0.20 SE).

Initial occupancy for models calculated at the two different spatial scales showed strong correlation (*r* = 0.87, *n* = 19, *p* < 0.01), suggesting that occupancy was similar from plot to landscape scales. Indeed, the spatial configuration of landscapes showed strong autocorrelation patterns at different spatial scales. This means that plots were normally embedded in a matrix of habitats of the same composition than the surrounding landscape (e.g., a plot placed in a forest showed similar vegetation profiles than the surrounding landscape; Figure 3). We calculated the occupancy for the 19 plots at the start and at the end of the study period (2007 and 2017) by fitting occupancy models to land use composition at the scale of 1 km × 1 km squares centred on the plots. In that case, we considered the first PC extracted considering the four main land uses (scrubland, forest, crops, and urban) showing a negative correlation with scrubland (*r* = −0.98, *p* < 0.0001, *n* = 850), positive correlation with forest (*r* = 0.86, *p* < 0.0001, *n* = 850), and no correlation with the two other land uses (PCAs with land uses in 2007 and 2017 were almost identical). These models considered that the landscape composition affected initial occupancy, colonization, extinction, and detectability. These probabilities of occupancy also showed strong correlation with the ones calculated at smaller spatial scales (500 m^−1^ km^2^: *r* = 0.90). Then, we fitted the initial probability of occupancy to the first PCA extracted with land uses calculated in 2007 and 2017 by using the best fitting regression model (third order polynomial regression: *r*^2^ = 0.998, df = 3, 15, *p* < 0.0001; *r*^2^ = 0.997, df = 3, 15, *p* < 0.0001, respectively, for years 2007 and 2017). Since the 19 plots were evenly distributed along the landscape gradient represented by the first PCAs (Figure 6a), and owing to the high model fits obtained, the predicted values were calculated for the entire study area, that is, the 850 1 km^2^ spatial units (Figure 6b). Predicted values were then projected to generate maps of occupancy for the years 2007 and 2017.

*C. russula* displayed significant differences in spatial and temporal patterns of probability of occupancy in the study area (Figure 7). The species showed higher occupancies in areas well covered by open land and lower occupancies in dense forested areas. Changes in occupancy between both periods were related to changes in habitat suitability along that period. In fact, some natural parks showed overall temporal decreasing habitat suitability (Montnegre, S. Litoral, and Garraf), with stronger scrubland encroachment (−8%) and afforestation (+6%). Other parks showed stability in habitat suitability (Sant Llorenç del Munt, Collserola, and S. Marina) due to moderate or not measurable landscape change. Occupancy change (e.g., the difference between predicted occupancy in both periods) depicted potential retractions of the spatial distribution of the species in the NE parks (−5% on average) but also in the SW (−0.5%), and potential expansions in the N (+4–6% in Sant Llorenç del Munt and Collserola). Differences in occupancy were mostly related to differences in habitat suitability associated to landscape change, which decreased in natural parks showing extreme afforestation processes (S. Litoral, Figure 8) and increased in parks showing moderate or no landscape change.

## 4. Discussion

This study highlighted the relevance of landscape change on the spatial and temporal distribution of *C. russula*. Heterogeneous responses of occupancy at the landscape scale were related to heterogeneous changes of land cover in different areas, producing significant changes in habitat suitability for the target species. Our results supported the key role of landscape change on the spatial distribution of this shrew, whose population dynamics were found to be little or not affected yet by ongoing climate change [14].

At the landscape scale (500 m-buffers), our study yielded consistent patterns of habitat change, with 14 out of 19 plots showing loss of open land cover and increase in forest cover, producing a significant decline of suitability for the target and other small mammal species dependent on open habitats (e.g., scrublands, [33]). Land cover change from abandoned open land to forests are representative of ongoing land use changes in protected areas of the study region [18] and were further confirmed in this study. Previous studies have associated recent landscape changes to shifts in the composition of small mammal communities over longer time periods (30 years, [44]), with a decrease of open land species and an increase of forest species, as was yet confirmed for other vertebrate taxa in the Mediterranean (birds, [45]). Results presented here would indicate faster changes in the spatial distribution of the target species than previously expected on the basis of climate change, driven by fast landscape changes.

Occupancy models predicted overall stability of the population during the studied period (except in pinewoods), which roughly coincided with the population size stability showed under a typical climate change scenario (e.g., increasing temperature and decreasing rainfall, [14]). These results suggested some resilience to climate change that was in contradiction to expectations for a small endothermic species with African origin [46]: if white-toothed shrews were constrained by cold temperatures, it would be expected of climate warming to produce positive effects on their populations. However, positive effects of climate change were neutralized by the more relevant and negative effects of landscape change. Indeed, shrews were constrained by two opposing driving forces: raising temperatures producing an increase in suitable potential areas (e.g., range expansions to the north and mountain areas [14]) and habitat loss (e.g., by afforestation) producing a decrease in suitable potential areas.

Predicted projections of the occupancy probability of *C. russula* in the whole area showed heterogeneous responses—with expansions and retractions of the species occupancy—exclusively based on main land cover changes. Expectations based on climate change using bioclimatic envelopes predicted moderate species retractions of the potential distribution in this region [47], highlighting that the use of appropriate spatial scales (smaller and adequate to the target species), altogether with the incorporation of biotic predictors in the models [48], will be necessary to have more real portraits of species ranges under global change scenarios. Indeed, the application of occupancy models allowed to predict changes on the spatial distribution of the species in such a large and heterogeneous area, owing to the predicted changes in habitat suitability during a relatively short period (10 years). The models showed that the greater white-toothed shrew displayed contrasting trends that shifted at relatively small spatial scales. Some actions aimed at the conservation of open land increased the favorability for shrews and many other related species [49]. Indeed, expanding ranges in the north could be associated to the successful application of policies related to the recovery of crops and management of forests affected by wildfires [19]. These actions produced that natural afforestation and scrubland encroachment processes were stopped or reversed, at least in some areas. On the other hand, natural parks from the NE were mostly unsuitable for shrews due to high forest cover and strong processes of afforestation and scrubland encroachment. Interestingly, in the SW, an area especially suitable for shrews owing to the high surface of scrubland available, we realized a predicted small loss of occupancy due to an overall decline in habitat suitability. Still, this area continues having one of the most abundant and persistent populations of shrews. Despite the predicted patterns of occupancy change were consistent with what can be expected owed to the habitat preferences of the shrew, we are aware about the relatively spatially reduced sample (n = 19 plots) used to make inferences on the occupancy for the whole study area. Moreover, we were concerned about the possibility of biased landscape changes at the periphery of the natural parks when changes occurred outside the protected areas. This can be important in small natural parks showing extensive boundaries with human-altered landscapes (e.g., urban areas and crops).

Theoretical models predicted direct relationships between abundance and habitat suitability [48], and our results confirmed that scrublands were high-quality habitats for shrews, holding higher mean abundance and occupancy, which in turn sustained populations with lower extinction rates and higher colonization rates [14,43]. On the other hand, broad-leaved forest showed the opposite, and in general, some population parameters declined along gradients of vegetation structural complexity from open to forest habitats. Nonetheless, habitat quality can be further affected by other landscape factors such as the isolation or connectedness of the habitat patches [10] that will surely affect dispersal and other population parameters under the source-sink dynamics typical of a metapopulation systems [48]. Indeed, our results indicated the particular case of pinewoods of *P. halepensis*, a habitat showing intermediate favorability between scrubland and broad-leaved forests regarding vegetation structure but suffering larger landscape change and fragmentation than the other plots. Being aware of the low sample size, the models predicted a decline of species occupancy due to low colonization rates caused by strong landscape change around pinewood plots. These pinewoods represent a special case of forest fragmentation caused by the opposite processes of land abandonment and wildfires [33,50,51], further influenced by strong edge effects and anthropogenic disturbances [52]. Small and isolated patches of pinewoods are almost the only representative forest habitat in the SW of the study area, probably producing a concentration of forest predators and competitors at higher rates than in continuous non-fragmented woodland of the NE.

Afforestation is a natural process resulting from the loss of traditional land uses and land abandonment in natural areas [53] and can be considered as one of the main conservation problems for biodiversity conservation in the Mediterranean basin. Nonetheless, afforestation and scrubland encroachment can counteract the effects of climate change by favoring more mesic microclimatic conditions under vegetation cover [54], and the rewilding process (e.g., restoring natural ecosystems processes through ecological succession, [55]) will have several additional benefits, such as the increase of bird populations associated to northern climates [45] and the increase of predators (birds of prey and carnivores, [55]). Negative effects on open-land species such as those documented here, and likely indirect effects linked to increased likelihood of large wildfires impossible to extinguish [8], can however counteract these positive effects. Fighting against this process is challenging and can be considered a wicked problem without a clear solution [56]. The management of protected areas is becoming more challenging with advancing climate change (and landscape change) and management techniques need to be adapted to particular current conditions [57]. However, reversing landscape change effects could be far beyond the power of the managers of Mediterranean protected areas, even considering the increasing impact of wildfires in the present context of climate change [18].

## Figures and Tables

**Figure 1 life-12-01230-f001:**
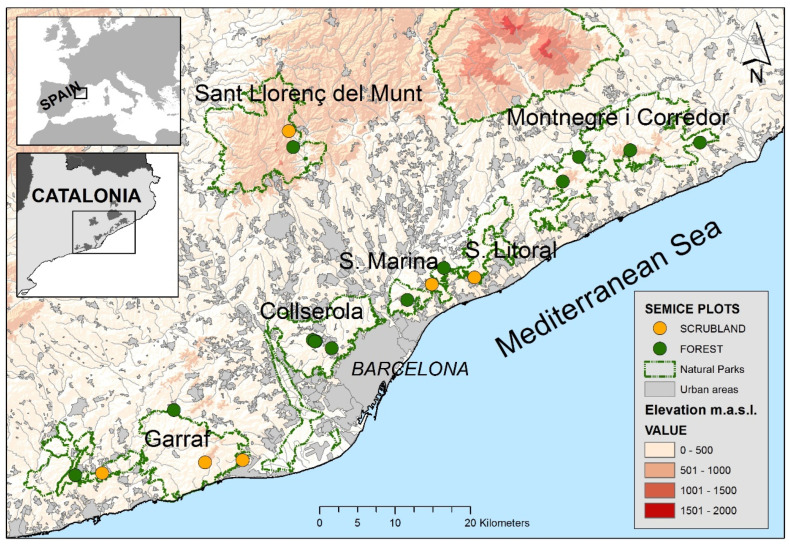
Situation of the 19 SEMICE sampling plots (13 forests and 6 scrublands) on the six natural parks of the Barcelona province (Catalonia, NE Spain).

**Figure 2 life-12-01230-f002:**
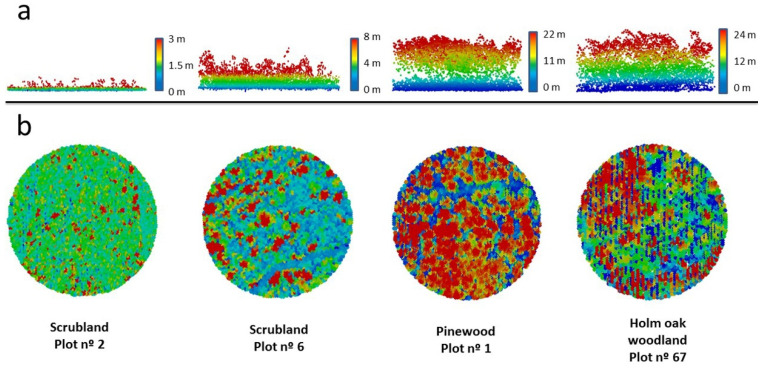
Example of vegetation profiles of four SEMICE sampling stations (two scrublands and two forests) calculated by LiDAR within a 50 mradius centered in the plots. Pixels ($\bar{X}$ = 1–4.3 per m^2^) are colored according to the height of vegetation (from dark blue to red) (**a**) Horizontal and (**b**) vertical views of the four plots.

**Figure 3 life-12-01230-f003:**
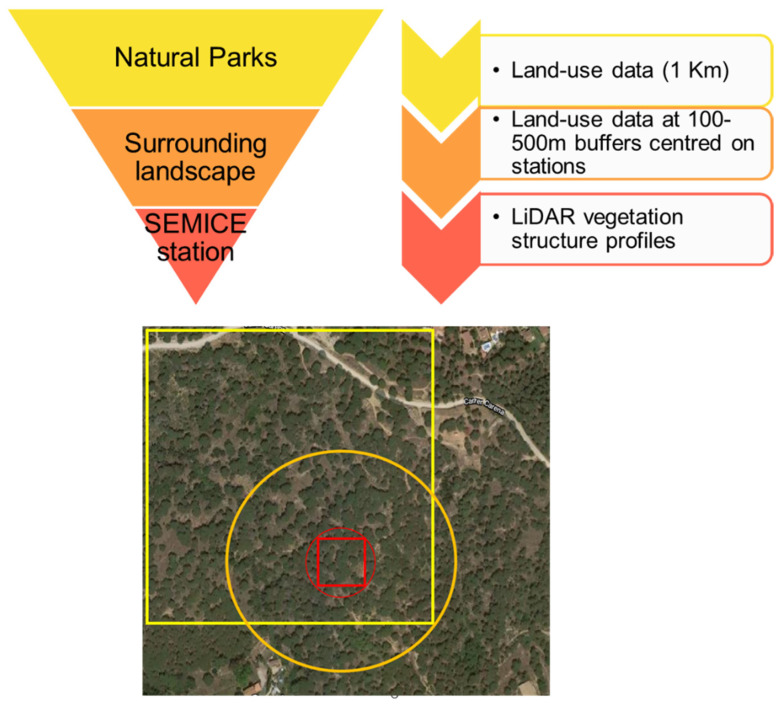
Scheme of the three-step process used to model the spatial distribution of *C. russula* in the study area. Red small figures indicate the plot level (square = SEMICE plot; circle = LiDAR calculations on 50 m radius), the orange circle was used for calculations of land-uses at the landscape scale (500 m buffer), and the yellow square indicates the land-use units used for the projection of models to the entire region. The rationale of this approach relies on the interdependence of habitats at the different spatial scales; that is, a plot placed in a forest showed similar vegetation profiles than the surrounding landscape, owed that forest patches are large enough and no fragmentation exists.

**Figure 4 life-12-01230-f004:**
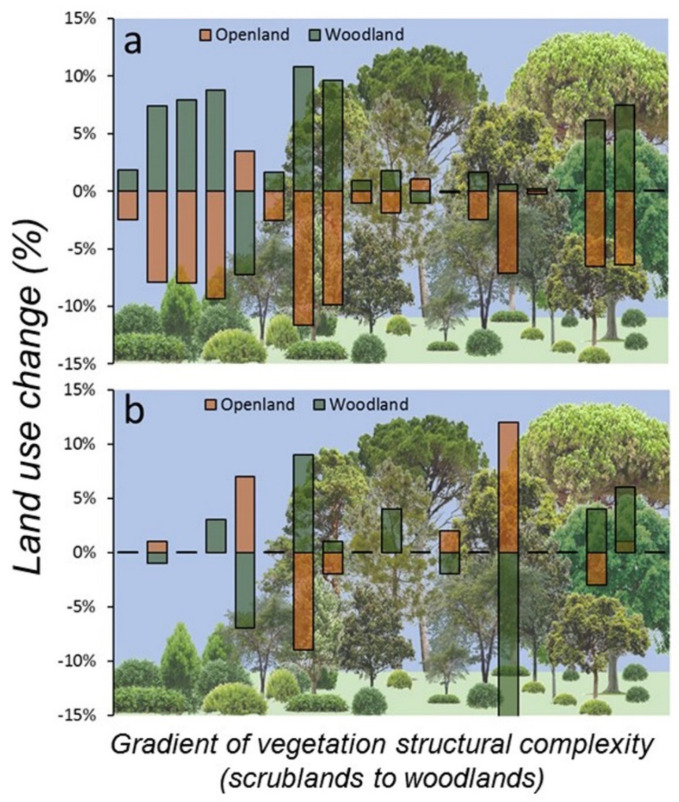
Landscape change (%), at two spatial scales (**a**) 500 and (**b**) 100 m buffers centred in the 19 SEMICE sampling plots in the period 2007–2017 for the two main structural habitats (scrubland-open land and forest-woodland) relevant for *C. russula* dynamics and demography in Mediterranean landscapes. Plots were ordered along the *x*-axis according to the factor scores of LiDAR-PC1, representing a gradient of vertical structural complexity of vegetation from scrubland to woodland (bottom figure).

**Figure 5 life-12-01230-f005:**
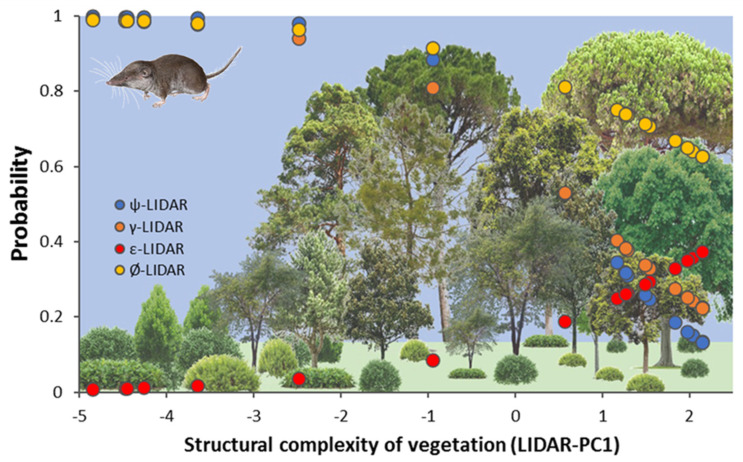
Graphical representation of the occupancy model selected using PRESENCE software indicating the changes in the modelled probability of four demographic parameters (occupancy-blue, colonization-orange, extinction-red, and persistence-yellow) along vegetation structural gradients represented by LiDAR-PC1 at the plot level. Bottom figure shows the changes in structural complexity of vegetation from scrublands to dense forests.

**Figure 6 life-12-01230-f006:**
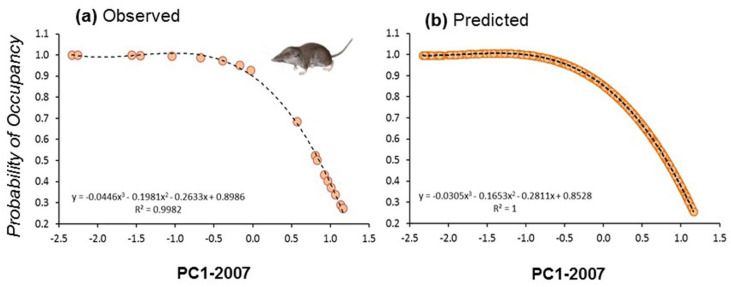
Modelled initial probability of occupancy of *C. russula* in 19 plots along a gradient of landscape composition represented by (**a**) observed values on the PC1-2007 (scrubland to woodland gradient) and (**b**) predicted values on PC1-2007 for the entire study area (850 km^2^). The same was calculated for 2017 (not shown).

**Figure 7 life-12-01230-f007:**
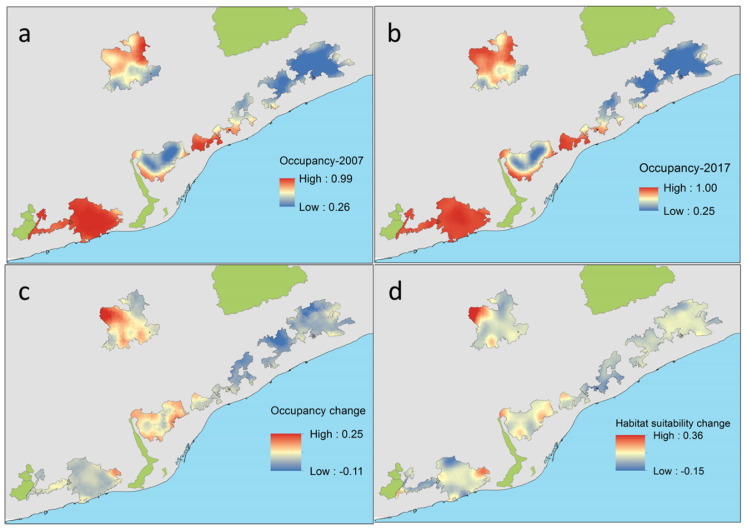
Predicted probability of occupancy of *C. russula* related to land cover uses (PC1) in the six natural parks of the study area at the start (2007, (**a**)) and at the end of the study (2017, (**b**)) and temporal changes in occupancy (**c**) and habitat suitability (**d**) between both periods (differences in rate of change for suitable-scrubland and crops-habitats). All values are probabilities.

**Figure 8 life-12-01230-f008:**
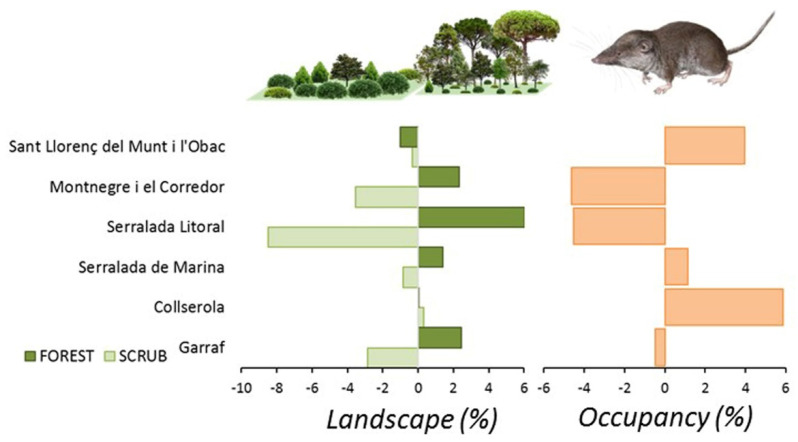
Landscape changes mostly affecting *C. russula* (scrubland encroachment and afforestation) and predicted relative changes in average occupancy in the six natural parks of the Barcelona province in the period 2007–2017.

**Table 1 life-12-01230-t001:** Land use changes (rate of change, %) in six natural parks of the Barcelona province in the period 2007–2017 and statistical significance after GLMs using rate of change of the main four land uses at each spatial unit (1 km^2^) as response variables and Natural parks as predictors. Sant Llorenç del Munt was set as the reference value.

NATURAL PARK	SCRUBLAND	WOODLAND	CROPS	URBAN
Garraf	−3.40 ***	2.37 ***	0.51	0.52 **
Collserola	0.00	−1.02	0.26	0.76 ***
Serralada de Marina	−2.18	0.96	−0.11	1.32 ***
Serralada Litoral	−8.58 ***	5.90 ***	1.58 *	1.10 ***
Montnegre i el Corredor	−3.58 ***	2.25 ***	0.80	0.53 *
Sant Llorenç del Munt i l’Obac	−0.20	−0.73	0.62	0.31
TOTAL AREA	−2.66 ***	1.33 ***	0.67 *	0.66 ***

* *p* < 0.05; ** *p* < 0.01; *** *p* < 0.001.

## Data Availability

The data presented in this study are available on request from the corresponding author.

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
