# Peer review of "Assessing the Effects of Landscape Change on the Occupancy Dynamics of the Greater White-Toothed Shrew Crocidura russula"

_life, 2022, doi:10.3390/life12081230_

Round 1

Reviewer 1 Report

The manuscript deals with an interesting issue of land change effects on the population dynamics of the shrew.  It is well written and based on solid data and methods. I appreciate that the authors approached the problem by analyzing both two dynamic phenomena (population dynamics and land use dynamics).

I suggest to edit few minor issues:

Fig. 1. Add names on national parks (and city of Barcelona) to the map for better orientation.

Subsection 2.3: List 12 mentioned variables in line 128

Lines 458-467: Shrubland enchroachment is a widely occurring phenomenon affecting formerly used (and now abandoned) areas. It can be seen as negative from the perspective of certain species, on the other hand, in a broader picture it could be viewed as a positive change towards more complex and stable habitats better adapted to ongoing climate change. Could authors maybe elaborate on this in the final chapter/conclusion?

Author Response

REV1

The manuscript deals with an interesting issue of land change effects on the population dynamics of the shrew.  It is well written and based on solid data and methods. I appreciate that the authors approached the problem by analyzing both two dynamic phenomena (population dynamics and land use dynamics).

I suggest to edit few minor issues:

Fig. 1. Add names on national parks (and city of Barcelona) to the map for better orientation.

Authors: We added the names of the Parks and Barcelona city

Subsection 2.3: List 12 mentioned variables in line 128

Authors: We included these variables in the Table S1, in which we also defined the meaning of each one and the way of calculating them.

Lines 458-467: Shrubland enchroachment is a widely occurring phenomenon affecting formerly used (and now abandoned) areas. It can be seen as negative from the perspective of certain species, on the other hand, in a broader picture it could be viewed as a positive change towards more complex and stable habitats better adapted to ongoing climate change. Could authors maybe elaborate on this in the final chapter/conclusion?

Authors: We have added the following paragraph: Afforestation and scrubland encroachment can counteract the effects of climate change by favoring more mesic microclimatic conditions under vegetation cover (De Frenne et al. 2019), and the rewilding process (e.g., restoring natural ecosystems processes through ecological succession, Navarro and Pereira 2015) will have several additional benefits, such as the increase of bird populations associated to northern climates (Seoane and Carrascal 2008) and the increase of predators (birds of prey and carnivores, Navarro and Pereira 2015). Negative effects on open-land species such as those documented here, and likely indirect effects linked to increased likelihood of large wildfires impossible to extinguish (Doblas_miranda et al. 2015), can however counteract these positive effects.

Reviewer 2 Report

Dear authors,

Your manuscript is very interesting, yet several issues need to be corrected prior to publication.

Figures – in every figure, where you have more than one picture or graph, you need to add a letter for this graph and describe it separately in the figure title. You did it in Figure 7, but please do it also in Figures 2, 4, and 6. Further, in the text, you should relate to these letters, so you don’t need to write Fig 6 – left or Fig 6-right as in lines 355-356.

Please add Summery, describing your main conclusions and the “take-home” message.

Lines 128-129 – you wrote about 12 variables from the LiDAR dataset that you use, without presenting these variables. Please add a new table, or direct the reader to Supplement S1, which presents these variables and their meaning.

Figure 2 – (1) please write the y-axis units in the upper graphs. (2) the lower small rectangles in plots 1 and 13 are not clear. Are these 3D pictures? I think you should remove it as I do not see how they contribute. Else, you should make it bigger, and add the same illustration to plots 2 and 6.

Author Response

REV2

Your manuscript is very interesting, yet several issues need to be corrected prior to publication.

Figures – in every figure, where you have more than one picture or graph, you need to add a letter for this graph and describe it separately in the figure title. You did it in Figure 7, but please do it also in Figures 2, 4, and 6. Further, in the text, you should relate to these letters, so you don’t need to write Fig 6 – left or Fig 6-right as in lines 355-356.

Authors: We changed the figures 2, 4 and 6, and the caption figures to match the new indications

Please add Summery, describing your main conclusions and the “take-home” message.

Authors: Done. We now explicitly say that landscape change may counteract climate change effects after controlling for land-use effects on local vegetation structure

Lines 128-129 – you wrote about 12 variables from the LiDAR dataset that you use, without presenting these variables. Please add a new table, or direct the reader to Supplement S1, which presents these variables and their meaning.

Authors: We included these variables in the Table S1, in which we also defined the meaning of each one and the way of calculating them.

Figure 2 – (1) please write the y-axis units in the upper graphs. (2) the lower small rectangles in plots 1 and 13 are not clear. Are these 3D pictures? I think you should remove it as I do not see how they contribute. Else, you should make it bigger, and add the same illustration to plots 2 and 6.

Authors: We have deleted the 3D figures, and only retained the horizontal and vertical views. We added “m” to the y-axes, and also made the caption clearer to the reader.

Reviewer 3 Report

Life - 1855169- Comments

This is a well written study that mapped the effects of landscape changes on the population distribution of C. russula. My suggestion for the title would be:

Assessing the effects of landscape changes on the population distribution of the greater

 white-toothed shrew Crocidura russula’

The authors should first adequately describe the Crocidura russula, in terms of its genus, habitat, reproduction, behavior etc… This is important especially because, the authors did not mention anything about the possibility of the presence of predators that may pray on the C. russula. For instance, considering that the species showed higher occupancies in areas well covered by open land and lower occupancies in dense forested areas. Could this not be due to presence of predators in the dense forested areas?

What were the limitations in this study? The authors need to mention them.

Minor comments:

·                  There some inconsistencies in the manner c. russula is written, please check throughout the document.

·                  What do the authors mean by the statement in line 101 since the sampling was not done by them? Please remove this statement as it is not necessary here and the mentioned fact cannot be guaranteed.

·                  What are the units for the grids in line 113?

·                  It is interesting that only the c. russula was captured during the trappings. I would expect other species to be captured unless this was a habitat occupied by only the c. russula. If other species were captured the authors need to mention them, since it is important to know which species were dominant in the area and if there were any territorial conflicts among them. This can also help us understand if the c.russula was not territorially dispossessed or preyed upon.

·                  What are the units of the vegetation profiles in figure 2?

·                  In line 185 to 191, the model formulation needs to be shown first before defining the parameters.

·                  In line 229, write ind./plot in full first and in line 245 GLM must be written in full first.

Author Response

REV3

This is a well written study that mapped the effects of landscape changes on the population distribution of C. russula. My suggestion for the title would be:

Assessing the effects of landscape changes on the population distribution of the greater white-toothed shrew Crocidura russula’

Authors: We agree, the original title did not match the actual contents of the paper, and we decided to combine your suggestion with ours: “Assessing the effects of landscape change on the occupancy dynamics of the greater white-toothed shrew Crocidura russula”. This title accords more with the fact that the paper analyses the dynamics of occupancy rather than changes in abundance, that is the first thing it comes into mind when speaking on population dynamics.

The authors should first adequately describe the Crocidura russula, in terms of its genus, habitat, reproduction, behavior etc

Authors: We have incorporated into the introduction short synthetic information on the biology and ecology of this shrew species.

This is important especially because, the authors did not mention anything about the possibility of the presence of predators that may pray on the C. russula. For instance, considering that the species showed higher occupancies in areas well covered by open land and lower occupancies in dense forested areas. Could this not be due to presence of predators in the dense forested areas?

Authors: We agree with your comments on predators’ effects on the spatial distribution of the shrew. In fact, in previous investigations, we have developed theoretical frameworks to explain the potential role of predators on the spatial distribution and abundance of small mammals along vegetation structural gradients (see refs. 36, 57). However, we believe that predators’ responses to landscape changes are less important at the temporal scales analyzed here. Nonetheless, we commented briefly on the role of observed landscape changes on the restructuration of communities (e.g., afforestation and rewilding processes).

What were the limitations in this study? The authors need to mention them.

Authors: We included one paragraph dealing with the possible limitations of our data. In particular, the relatively spatially reduced sample (n = 19 plots) and the inference performed to predict the occupancy for the whole study area. Also, we were aware about the possibility of biased landscape changes at the periphery of the Natural Parks when changes occurred outside the protected areas. This can be important in small parks with boundaries in contact with human-altered landscapes (e.g., urban areas and crops).   

Minor comments:

  • There some inconsistencies in the manner c. russulais written, please check throughout the document.

Authors: We checked inconsistencies and corrected them.

  • What do the authors mean by the statement in line 101 since the sampling was not done by them? Please remove this statement as it is not necessary here and the mentioned fact cannot be guaranteed.

Authors: We deleted this statement since we have no proof.

  • What are the units for the grids in line 113?

Authors: We have included the size of the grids adding the distance between traps.

 It is interesting that only the c. russula was captured during the trappings. I would expect other species to be captured unless this was a habitat occupied by only the c. russula. If other species were captured the authors need to mention them, since it is important to know which species were dominant in the area and if there were any territorial conflicts among them. This can also help us understand if the c.russula was not territorially dispossessed or preyed upon.

Authors: We captured other small mammal species, but to be brief we omitted that information. We included a sentence with the other species trapped in order to have a more real interpretation of small mammal communities at the studied plots.

  • What are the units of the vegetation profiles in figure 2?

Authors: Also raised by rev.2, we included the units in the y-axis (m) and further explained the meaning of the colors.

  • In line 185 to 191, the model formulation needs to be shown first before defining the parameters.

Authors: Done

  • In line 229, write ind./plot in full first and in line 245 GLM must be written in full first.

Authors: Done

Regarding excessive self-citations: The fact is that we have been studying the study system (small mammals in Mediterranean forest and scrublands) during the last dècades so we had to refer frequently to former publications to emphasize what is new and what results are related to previous ones. Further, we have also worked recently on wider-scale papers dealing with the utility of studies of this sort for understanding and managing Mediterranean systems in the face of ongoing land use changes. This explains the comparatively large numbers of our own papers we cite. Anyway, we have now deleted 5 out of the 15 (from a total of 61) self-citations, the ones less directly related to the topics addressed. We hope this will suffice.
